# Perinatal factors affect the gut microbiota up to four years after birth

Fiona Fouhy [1,2], Claire Watkins[1,2], Cian J. Hill[1], Carol-Anne O'Shea[3], Brid Nagle[2], Eugene M. Dempsey[3,4], Paul W. O'Toole[1,5], R. Paul Ross[1,5], C. Anthony Ryan[1,3] & Catherine Stanton[1,2]

Perinatal factors impact gut microbiota development in early life, however, little is known on the effects of these factors on microbes in later life. Here we sequence DNA from faecal samples of children over the first four years and reveal a perpetual evolution of the gut microbiota during this period. The significant impact of gestational age at birth and delivery mode on gut microbiota progression is evident in the first four years of life, while no measurable effects of antibiotics are found in the first year. Microbiota profiles are also characteristic in children dependant on gestational age and maturity. Full term delivery is characterised by *Bacteroides* (year one), *Parabacteroides* (year two) and *Christensenellaceae* (year four). Preterm delivery is characterised by *Lactobacillus* (year one), *Streptococcus* (year two) and *Carnobacterium* (year four). This study reveals that the gut retains distinct microbial profiles of perinatal factors up to four years of age.

[1] APC Microbiome Ireland, Cork T12 YT20, Ireland. [2] Teagasc Food Research Centre, Moorepark, Fermoy, Co, Cork P61 C996, Ireland. [3] Department of Neonatology, Cork University Maternity Hospital, Cork T12 YE02, Ireland. [4] INFANT Centre, University College Cork, Cork T12 YT20, Ireland. [5] School of Microbiology, University College Cork, Cork T12 YT20, Ireland. These authors contributed equally: Fiona Fouhy, Claire Watkins. Correspondence and requests for materials should be addressed to C.S. (email: catherine.stanton@teagasc.ie)

The gut microbiota impacts on a range of host biological processes, such as host metabolism, immune defence and cognitive neurodevelopment[1–4]. The infant gut microbiota is significantly affected by factors such as mode of delivery and gestational age at birth[5–7] until 2–3 years of age when it is thought to resemble a more stable adult-like microbiota[8,9], though the exact timing of gut microbiota stabilisation (maturation) remains unclear. Previously, we reported on the characterisation of gut microbiota development during the first 24 weeks in a cohort of initially breastfed infants born full term (FT) and preterm (PT) (<35 weeks gestation) via vaginal delivery and Caesarean section (CS) birth modes[10]. We demonstrated that mode of delivery, gestational age at birth and breastfeeding had significant effects on the microbiota composition over the first 24 weeks.

Infants born prematurely (<35 weeks gestation) have a gut microbiota with reduced diversity and lower levels of *Bifidobacterium* and *Bacteroides* compared to FT infants[11,12]. Worldwide approximately 10% of babies are born prematurely and up to 25% of PT survivors have adverse neurodevelopmental outcomes[13]. In addition, infants born prematurely are likely to receive antibiotic treatment, which exerts significant effects on gut microbiota[14]. The effects of prematurity are often further confounded by CS delivery, with 43% of PT and 67% of very PT infants born by CS[15]. In terms of progression, patterns of microbial colonisation in the infant gut have primarily been associated with gestational age at birth, after adjusting for antibiotic exposure, mode of delivery and breastfeeding status, among others[12,16,17]. Indeed, many of these PT studies include very low birth weight (VLBW) infants with extended hospital care, thereby exposing the juvenile microbiome to surfaces in the neonatal intensive care unit, which have previously been shown to influence gut microbiota colonisation[18]. Birth weight is therefore an underlying factor that plays a role in microbiota progression as VLBW infants also possess an immature immune system that influences microbe–gut interactions[19]. Furthermore, the convergence of gut microbiota profiles has been associated with increases in *Bifidobacterium* spp. in FT infants between ~60 and 130 days post birth[20]. Research has shown that PT infants have an altered gut microbiota compared to FT infants in early life[10], though limited research has been reported on later development of their gut microbiota, which has recently been highlighted as a research priority[19].

Delivery mode has been shown to significantly impact on the gut microbiota. Infants born by CS are colonised with greater numbers of microbes from skin and environmental origins, than vaginally delivered infants[21,22]. Furthermore, studies have shown that those born by elective CS have particularly low richness and diversity[23]. Feeding regime also influences the acquisition and establishment of the infant gut microbiota. Breastfed infants are reportedly colonised with microbes present in breast milk and with greater numbers of *Bifidobacterium* compared to formula-fed infants[24–26]. We have previously shown that significant differences in gut microbiota were apparent when comparing CS infants (but not vaginally born infants) who were breastfed for <4 months compared to >4 months[10].

Thus, while extensive research exists on the infant gut microbiota establishment within the initial days, weeks and months of life, there remains limited data on the gut microbiota later in life. In this study, we show that perinatal factors including delivery mode and gestational age at birth result in distinct microbial profiles extending to four years of age.

## Results

**Participant characteristics.** Faecal samples were collected from individuals at years one ($n = 70$), two ($n = 57$) and four ($n = 32$) (all of whom were previously studied as part of the INFANTMET study up to 24 weeks of age) (Supplementary data 1). Of the 70 participants at year one, 47% were studied at year two and 20% at year four. Nine percent of the study participants were studied at all three time points. With respect to gestational age at birth, 87% (61/70) of infants studied at year one were born FT, 89% (51/57) at year two and 75% (24/32) at year four. Of the PT born individuals studied, 6% were extremely low birth weight (<1000 g), 19% were VLBW (<1500 g), 62% were low birth weight (<2500 g) and 13% were >2500 g. Metadata including details of delivery mode, gestational age at birth, breastfeeding duration and antibiotic exposure are provided in Supplementary data 2.

**Gut microbiota diversity increases with age.** The 16S rRNA sequencing resulted in >37,000 sequencing reads per sample, with no significant differences between the number of sequencing reads per samples at year one, two or four, (Supplementary Fig. 1).

Using a redundancy analysis (RDA) plot to explore complex associations between community composition and multiple explanatory variables, we determined that age ($p \leq 0.001$) and gestational age at birth ($p < 0.05$; RDA) had a significant impact on the microbiota (Fig. 1a). Similarly, using canonical correspondence analysis (CCA), we determined that age ($p \leq 0.001$; CCA) and gestational age at birth ($p < 0.01$; CCA) had a significant impact on the microbiota (Fig. 1b). Alpha diversity significantly increased with age based on several measures, including Shannon diversity and evenness (Fig. 2a). The highest alpha diversity was found in those born FT ($n = 136$), with lower diversity in those born PT ($n = 23$), when all samples from years one, two and four, were included (Fig. 2b). Beta diversity measures the between-group differences in diversity. Based on Bray–Curtis distance matrices on data from individuals sampled at one time point (year one, two or four), there was a notable separation based on age up to four years (Fig. 3a, c). The strongest significant effect was identified between years one and four, with the highest $R^2$ value of 0.176 and $p < 0.001$ (Adonis) (Fig. 3a, c). Gestational age at birth also has a significant impact on community composition based on operational taxonomic unit (OTU) variance using Bray–Curtis distance matrices ($p < 0.01$; Adonis) (Fig. 3b, c). No visible clustering of samples occurred based on delivery mode, duration of breastfeeding, exposure to antibiotics/probiotics in the first year of life and the presence of siblings in a household (Supplementary Fig. 2A–E).

To further interrogate the factors associated with the clustering of samples based on diversity, we used principal component analysis (PCA+) to map gestational age at birth and current age onto the gut microbiota data (Fig. 4). It was evident at year one that a separation occurred between PT ($n = 9$) and FT ($n = 61$) children. In addition, PT children at year four ($n = 8$) clustered with FT children at year 2 ($n = 51$), suggesting a delayed progression in microbiota diversity in those born PT. PERMDISP2 permutational analysis, which tests whether community composition is significantly different between groups based on the distances of each sample to the group centroid in a principal coordinate analysis, showed separation of samples at years one, two and four (Supplementary Fig. 3).

A heatmap was used to determine patterns in gut microbiota based on age of the participant. Genera highlighted on the left-hand side of the map (C), including *Escherichia–Shigella* and *Enterobacter* were present at greater abundances at year one, whereas genera highlighted on the right-hand side of the map (A, B and D), including *Ruminococcaceae* spp. and *Christensenellaceae* spp., were present at greater abundances and *Flavonifracter*

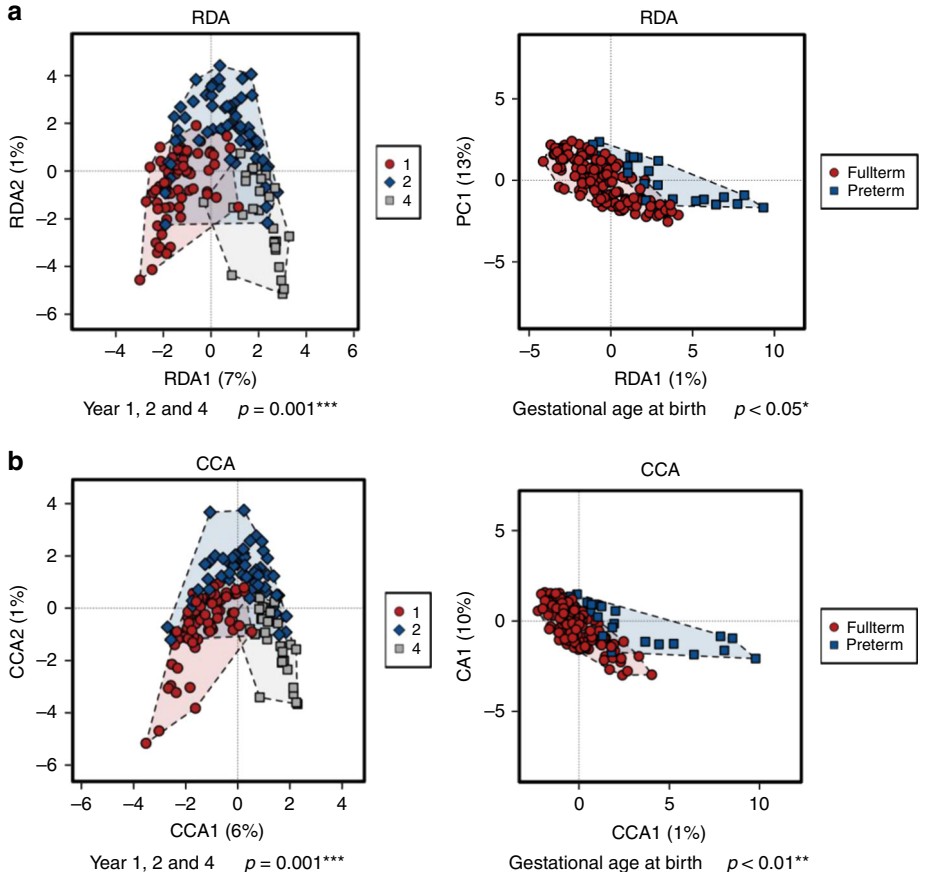

**Fig. 1** Samples separate based on year and gestational age at birth. **a** Redundancy analysis (RDA) of operational taxonomic units (OTUs) (top 1000 OTUs present) showing separation of samples by year. **b** Canonical correspondence analysis (CCA) on OTUs, with samples shown at year one, two and four. Source data are provided as a Source Data file. (*$p < 0.05$, **$p < 0.01$, ***$p < 0.001$)

and *Veillonella* spp. were present at lower abundances at years two and four (Fig. 5).

**Gut microbiota composition at years one, two and four of life**. To investigate the changes in the gut microbiota of children between years one, two and four of life, we conducted repeated-measures statistical analysis on the data. Participants who were sampled at more than one time point were included in the analysis, i.e. 27 individuals gave samples at years one and two, with six individuals giving samples at all three time points (Supplementary data 2). At the phylum level, *Bacteroidetes*, *Verrucomicrobia* and *Actinobacteria* were all significantly different in abundance between time points ($p < 0.05$, repeated-measures statistical analysis). At the family level, there were 24 significantly different taxa found between time points, including *Porphoromonadaceae* ($p < 0.001$, repeated-measures statistical analysis), *Bifidobacteriaceae* ($p < 0.05$, repeated-measures statistical analysis), *Bacteroidaceae* ($p < 0.05$, repeated-measures statistical analysis), *Acidaminococcaceae* ($p < 0.05$, repeated-measures statistical analysis), *Peptostreptococcaceae* ($p < 0.05$, repeated-measures statistical analysis) and *Lactobacillaceae* ($p < 0.05$, repeated-measures statistical analysis). At the genus level, there were 56 significantly different genera found between time points, including *Sutterella* ($p < 0.01$, repeated-measures statistical analysis), *Faecalibacterium* ($p < 0.05$, repeated-measures statistical analysis) *Parabacteroides* ($p < 0.05$, repeated-measures statistical analysis), *Ruminiclostridium* spp. ($p < 0.05$, repeated-measures statistical analysis) and *Lachnospiraceae* ($p < 0.05$, repeated-measures statistical analysis) (Supplementary data 3).

**Identifying discriminative gut taxa at years one, two and four**. To identify the most discriminative taxa/OTUs that best characterise microbiota composition at years one, two and four, sparse partial least squared–discriminative analysis (sPLS-DA) was conducted using a repeated-measures design on the top 1000 most abundant genera present at ≥2 time points (Fig. 6a). *Enterobacter*, *Haemophilus* and *Bifidobacterium* were found to best characterise the microbiota profile at one year, *Holdemania*, *Ruminococcaceae* UCG005 and *Christensenellaceae* R7 at two years and *Ruminococcaceae* UCG004 and *Coprococcus_1* at four years (Fig. 6b).

**Gestational age impacts gut microbiota up to four years of age**. In order to determine whether gestational age at birth exerted measurable effects on gut microbiota at one, two and four years of life, ANalysis of COmposition of Microbiomes (ANCOM) was performed on the top 1000 most abundant taxa at the phylum, family and genus levels. At year one, no significant differences occurred at the phylum level between those born FT versus PT. At the family level, *Alcaligenaceae* were only detected in FT born infants ($p < 0.05$, ANCOM) and at the genus level *Tyzerella_4* was significantly higher in those born PT compared to those who were born FT ($p < 0.05$, ANCOM). To determine whether there were bacteria present in the gut microbiota at one year that could discriminate based on gestational age at birth, we used a feature selection statistical analysis (LDA Effect Size (LEfSe)), which determines the features (in this study, genera) most likely to explain differences between groups. There were eight genera that discriminated those who had been born PT from those born FT,

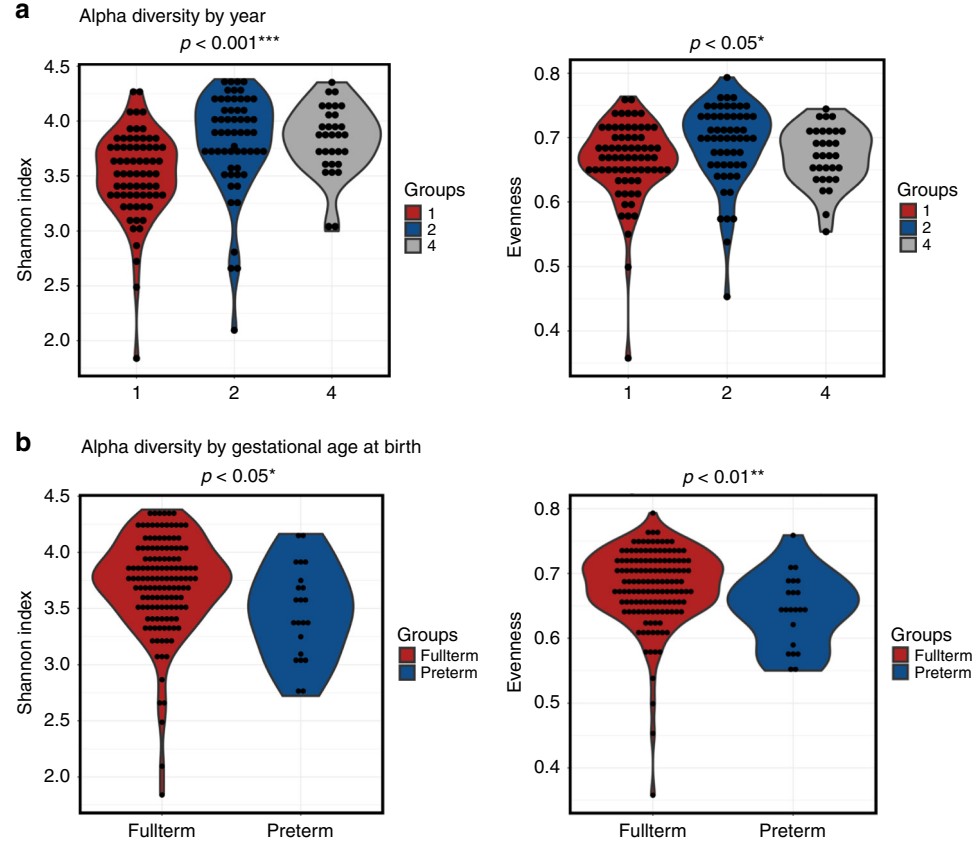

**Fig. 2** Alpha diversity increases with age. Alpha diversity shown as Shannon Index and Evenness based on **a** year of sample and **b** gestational age at birth. Source data are provided as a Source Data file. (*$p < 0.05$, **$p < 0.01$, ***$p < 0.001$)

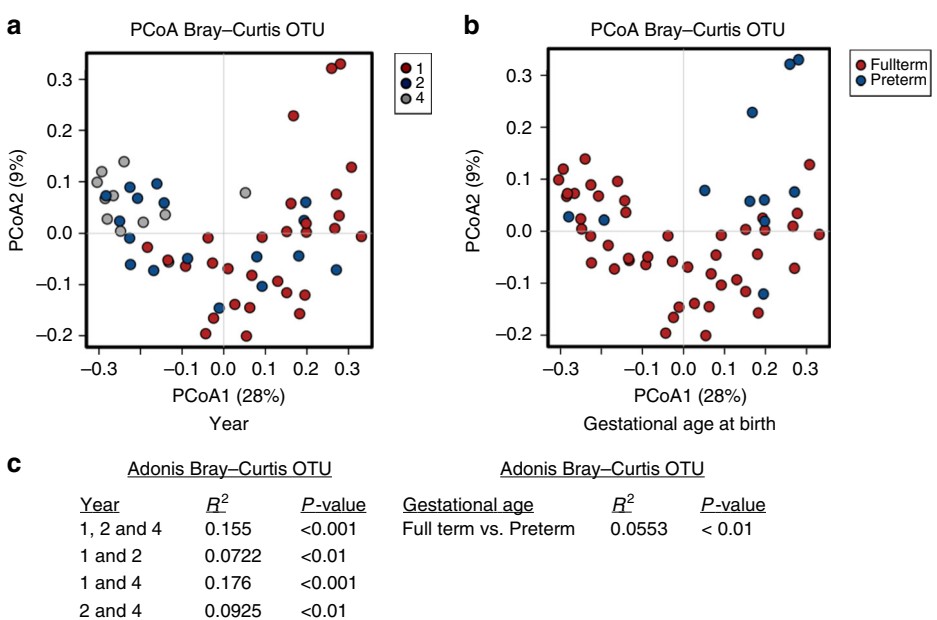

**Fig. 3** Separation of samples using beta diversity data based on age and gestational age at birth. **a** Principal coordinate analysis (PCoA) based on Bray-Curtis operational taxonomic unit (OTU) data on independent samples at years one, two and four (individuals sampled at one time point). **b** PCoA based on Bray-Curtis OTU data on independent samples based on gestational age at birth. **c** Adonis variance analysis based on Bray-Curtis distance matrices at OTU level. Source data are provided as a Source Data file. (*$p < 0.05$, **$p < 0.01$, ***$p < 0.001$)

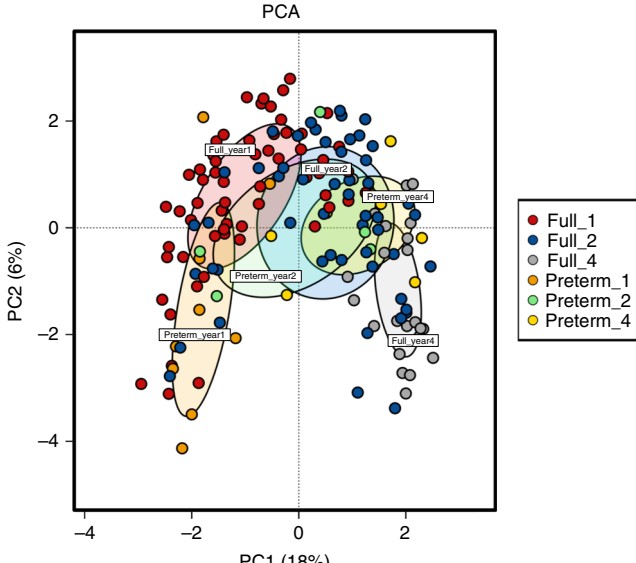

**Fig. 4** Factors influencing separation of samples based on beta diversity PCA+ using operational taxonomic units (OTUs; top 1000 OTUs) and metadata from infants at years one, two and four. Source data are provided as a Source Data file

at one year of age, with *Lactobacillus* having the greatest discriminatory power for PT delivery. FT birth was discriminated by higher *Bacteroides* and *Faecalibacterium*, *Veillonella*, *Sutterella*, *Parabacteroides* and various *Lachnospiracheae* and *Ruminococcaceae* spp. (Supplementary Fig. 4A).

At two years of age, there were no significant differences between children based on gestational age at birth, at the phylum and family levels. At the genus level, *Parabacteroides* were only detected in the FT delivered infants at year two. Using LEfSe analysis, several discriminative bacteria were identified at year two based on gestational age with *Streptococcus* having the greatest discriminatory power for PT delivery, while *Parabacteroides* was associated with FT delivery (Supplementary Fig. 4B).

At four years of age, there were no significant differences between the gut microbiota at the phylum level of children based on gestational age at birth. At the family level, FT delivery resulted in significantly increased *Clostridiaceae* and, at the genus level, significantly higher *Coprococcus*. Discriminative bacteria were identified between children based on gestational age at birth including *Carnobacterium*, *Desulfovibrio* and *Pharscolaractobacterium* in PT born children, while *Christensenellaceae* and various *Ruminococcaceae* and *Lachnospiraceae* spp. discriminated those who had been born FT (Supplementary Fig. 4C).

**Delivery mode impacts on gut microbiota up to four years**. In order to determine whether delivery mode exerted measurable effects on gut microbiota at one, two and four years of life, ANCOM was used. At one year, there were no significant differences at the phylum level between children based on delivery mode. At the family level, *Porphoromonadaceae* were significantly higher in those who had been born vaginally ($p < 0.01$, ANCOM), and at the genus level these individuals also had higher levels of uncultured organisms ($p < 0.05$, ANCOM). To determine the discriminative bacteria based on delivery mode at year one, LEfSe analysis was conducted. This analysis determined *Sutterella*, *Parabacteroides* and uncultured organisms to be the discriminative bacteria of vaginally delivered individuals at

year one, while *Akkermansia*, *Citrobacter*, *Lachnospiracheae* UCG008 and *Coprococcus* 2 were the microbes that discriminated CS delivered children at one year (Supplementary Fig. 5A).

When the gut microbiota of two-year-old children was studied, no significant differences were determined using ANCOM between those born vaginally or by CS. LEfSe identified *Parabacteroides* and *Ruminiclostridium* as the discriminating bacteria for vaginally delivered children, while *Gordonibacter* and *Lachnospiracheae* NC2004 group were discriminative of CS delivery, based on gut microbiota two years after birth (Supplementary Fig. 5B).

At four years, there were no significant differences detected in the gut microbiota based on delivery mode, at the phylum or family levels. At the genus level, vaginally delivered individuals had significantly higher uncultured organisms compared to those delivered through CS. Six genera were identified as discriminatory for vaginal delivery based on the gut microbiota at four years, namely *Coprococcus* 2, *Ruminiclostridium* 1, *Lachnospiraceae* UCG001, *Clostridium sensu strictu* 1, *Peptoclostridium* and uncultured organisms (Supplementary Fig. 5C).

## Discussion

Extensive research now exists around the acquisition and development of the infant gut microbiota during delivery and in the initial weeks of life. However, research determining how long early-life factors continue to exert measurable effects on gut microbiota remains limited[5,8]. Thus this prospective study investigated whether perinatal factors including gestational age at birth and delivery mode would be remembered by the gut microbiota, resulting in distinct microbiota profiles up to four years of age.

In line with the existing data, our study confirms the continued development and maturation of the gut microbiota during the initial years of life. We identified an increase in gut microbiota diversity with increasing age to four years, with the greatest significant difference found between infants at year one and year four. This continuous shift in microbiota development was previously described in a Danish cohort of infants from nine to 36 months of age[27]. This gradual change in microbiota diversity has been associated with diet and other environmental factors previously described[28,29]. Conversely, factors such as antibiotic exposure have been shown to reduce gut microbiota diversity. In addition, lower diversity has been associated with later adverse health outcomes, including atopy and allergic diseases[30–32].

The abundance of several phyla differed significantly with the age of the participant. *Bacteroidetes* was found to significantly decrease with age in children born FT; however, the opposite occurred in those born PT where *Bacteroidetes* significantly increased during the first four years of life. *Actinobacteria* was found to significantly decrease over time in all children (born FT or PT). These results are not surprising as *Actinobacteria* are known to decrease with age, as soon as breastfeeding has ceased and weaning has begun[29,33]. Furthermore, a number of genera were found to significantly decrease over time in the FT subjects, including *Bacteroides* and *Bifidobacterium*. Genera found to increase significantly over time in the FT cohort included *Coprococcus*, *Lachnospiraceae* and *Dorea* (repeated-measures analysis), previously associated with an adult-like microbiota[29,34]. Yatsunenko et al. examined the gut microbiome of healthy children and adults from the Amazonas of Venezuela, rural Malawi, and USA metropolitan areas and found that *Bifidobacterium* species were dominant throughout the first year of life and continued to decrease during this period[5]. Interestingly, the children at year four in the current study had a gut microbiota profile dominated by taxa such as *Ruminococcaceae*, *Dialister*,

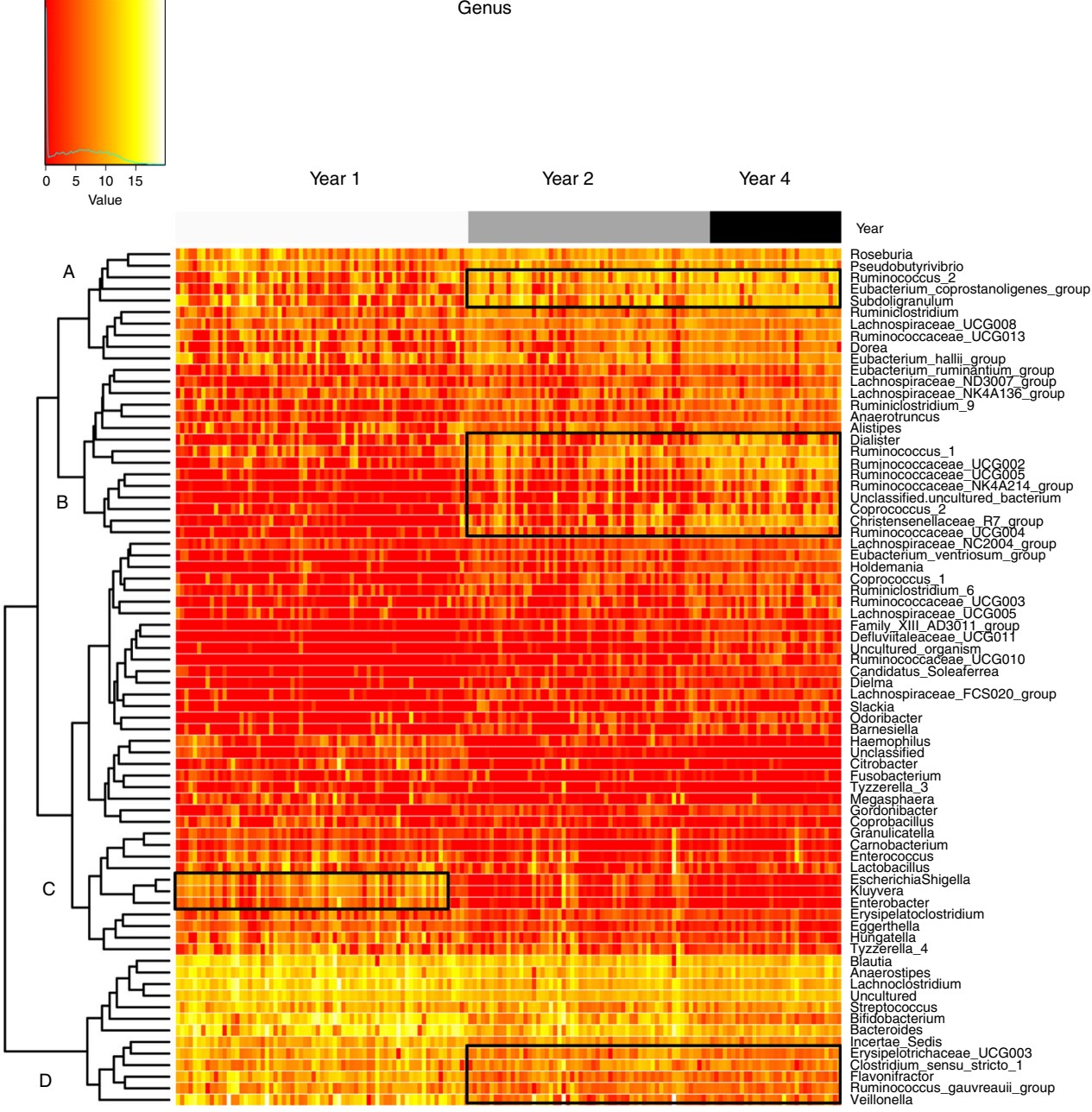

**Fig. 5** Microbiota associated with the age of the infant. Heatmap based on top 100 genera clustered based on year. Values range from low (red) to high (yellow). Source data are provided as a Source Data file

*Faecalibacterium*, *Bacteroides* and *Christensenellaceae*. Long-term dietary interventions have found *Bacteroides* associated with protein and animal fat intake, thus the microbiota likely reflect adaptation to the increasingly adult-type diet by four years[35].

Similar in-depth analysis of the infant gut microbiome from birth to three years of age has previously been described, whereby shifts in microbiota composition were predominantly altered by life events, such as dietary changes, antibiotic administration and illness[6,8,36]. Our results further expand these previous findings and demonstrate that the gut microbiota continues to evolve up to four years of age.

In addition to the impact of age on gut microbiota diversity, we also investigated whether perinatal factors including gestational age leave distinct microbial profiles on the microbiota up to four years of age. The results showed that gestational age at birth and delivery mode continued to impact diversity during the first

four years of life with those born FT having the highest diversity at one, two and four years of age, while PT infants had lower diversity. As many of our PT infants were born by CS, this reduced diversity is not surprising. Our previous research on these infants up to 24 weeks also demonstrated that alpha diversity continues to increase from birth to 24 weeks of age, but the diversity did not increase evenly, with FT vaginally delivered infants having the highest diversity, and PT vaginally delivered infants having the lowest diversity[10].

Based on gestational age at birth, certain microbiota were capable of discriminating between children up to four years. At one year, *Bacteroides* and *Faecalibacterium* best discriminated FT born individuals, while *Lactobacillus* was the discriminatory bacteria for PT delivered individuals. At two years, *Streptococcus* was discriminatory of PT delivery, while *Parabacteroides* was discriminatory of FT birth. By four years, *Christencenellaceae* and

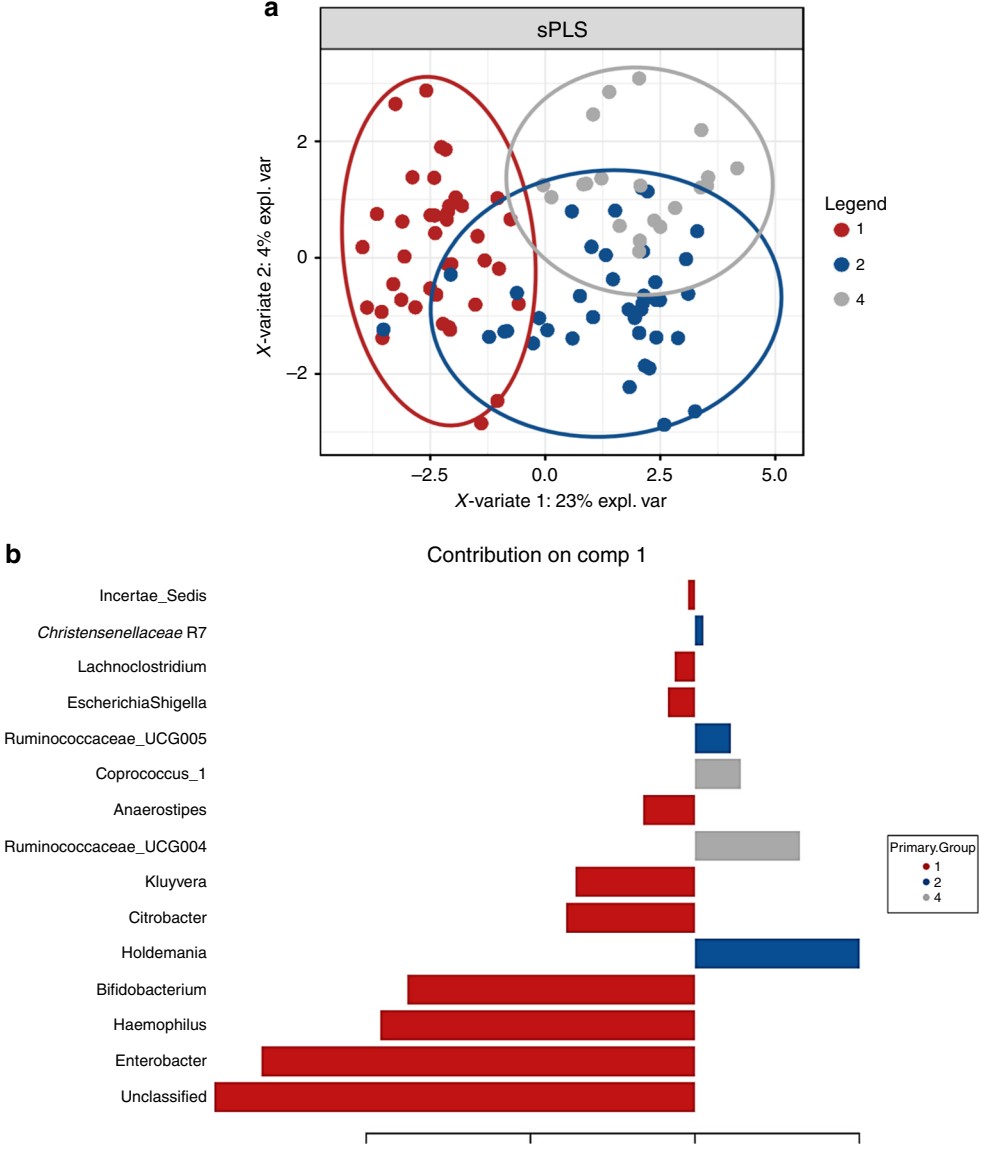

**Fig. 6** Taxa associated with the age of the individual. **a** Sparse Partial Least Squared–Discriminative Analysis plot illustrating a clear separation on repeated measures between years one, two and four using the top 1000 most abundant genera. **b** The associated contribution plot illustrating taxa associated with each year. Source data are provided as a Source Data file

*Ruminococcaceae* best discriminated FT delivered individuals, while *Carnobacterium* was associated with PT delivery. Previous studies have found a delayed succession of microbial species in PT infants[16], which may be attributed to gestational age at birth and antibiotic exposure[37–39]. There is currently a paucity of data surrounding PT infant gut microbiota acquisition and development and its impact on later health outcomes, thus this study provides important insights into the impact of gestational age at birth on gut microbiota development up to four years of age. The abundance of *Lactobacillus* at year one in PT children may reflect a less adult-like gut microbiota, compared to FT infants. Indeed, Chernikova et al. found that gestational age had a positive effect on the abundance of *Lactobacillus* in PT infants[37] and increased levels of *Lactobacillus* were previously reported in PT infants compared to FT infants at three months of age[40].

With a global increase in CS deliveries[41], we were interested in studying the longer-term impact of delivery mode on gut microbiota development over the first four years of life. Our study found that *Lachnospiraceae* was dominant in CS delivered

children and *Parabacteroides* was found to be more unique to vaginally delivered children at one and two years of age; however, by year four, *Lachnospiraceae* and other *Clostridium* spp. became more dominant in these vaginally delivered children. Recent findings have further demonstrated the continued impact of perinatal factors, including delivery mode, on gut microbiota succession[42]. Thus our study further demonstrates the continued impact of key perinatal factors on gut microbiota development and the necessity for further studies to interrogate the associated health impacts.

In our previous study on these infants up to 24 weeks, we demonstrated the significant effect of breastfeeding duration on the microbiota of the FT CS infants but not on the FT vaginally delivered infants[10]. Five genera were found to be present at significantly different abundances at week 24 due to prolonged breastfeeding (>4 months): *Sutterella, Clostridium*_XIV, *Megamonas, Lachnoanaerobaculum,* and *Megasphaera.* However, in this follow-up study, we did not detect significant differences in gut microbiota at years one, two or four due to breastfeeding.

This is likely due to the fact that the early-life impact of breast-feeding can no longer be detected, perhaps due to the influence of other factors, including diet, age and environment.

No sustained effects of antibiotic exposure during the first year of life were observed in this study. However, this may reflect a lack of detailed longitudinal antibiotic data in this study. We have previously shown that early-life exposure to antibiotics results in alterations to the microbiota, with incomplete recovery up to two months post treatment, with others showing recovery of the infant gut microbiota following short-term antibiotic use[6,14]. In addition, we did not observe significant effects of antibiotic exposure on diversity measures across the first four years of life. A recent study demonstrated that antibiotic-treated children showed less diversity at the species and strain levels in their gut microbiota and some species were dominated by a single strain over the first 36 months of life[6]. In addition, that study also showed that the ability of antibiotic resistance genes on mobile elements persisted longer after antibiotic exposure and that the gut microbiota of antibiotic-treated children was less stable compared to antibiotic-naive infants. Fewer studies have examined the impact of antibiotics on the gut microbiota beyond one year of age. A study of 42 Finnish children found that macrolide use resulted in a long-term reduction in gut microbiota richness that persisted even 24 months later[43]. Further studies applying shotgun sequencing coupled with detailed antibiotic history are required to increase our understanding of the impact of early-life exposure to antibiotics on gut microbiota composition, stability and the development of a resistance reservoir.

This study has some limitations. Firstly, in this study we were unable to examine the impact of prolonged breastfeeding on gut microbiota. While we recorded breastfeeding duration as less than or greater than four months, we could not determine whether prolonged breastfeeding resulted in a beneficial effect on gut microbiota up to four years of age. Secondly, this study has a relatively small number of PT participants ($n = 16$); of the PTs studied, only 6% were extremely low birth weight (<1000 g), 19% were VLBW (<1500 g) and 13% were >2500 g. However, we have shown that gestational age is a more accurate measure of the impact of prematurity, as not all low birth weight infants are PT. Furthermore, this study is, to our knowledge, the longest newborn microbiota study to date.

The results of this study have shown that gestational age at birth still imprints on the microbiome at four years of age. This is extremely important as now worldwide >10% of babies are born prematurely annually and up to 25% of PT survivors are left with adverse neurodevelopmental outcomes. This study is limited to correlations rather than causation between prematurity and gut microbiota development and thus we cannot predict the implications on neurodevelopmental outcomes of the altered microbiome in prematurity. Our data reveal an association between prematurity and a persistent microbiome imprint at least up to four years of age, highlighting the opportunities for large-scale studies to investigate the relationship between gestational age, gut microbiota and the gut–brain axis. Moreover, this study opens up opportunities whereby live biotherapeutic interventions may be beneficial in PT infants to modulate the gut microbiota to resemble their FT born counterparts.

## Methods

**Participants and sample collection.** The participants in this study were recruited as part of the INFANTMET study cohort, based in Cork, Ireland[10]. Ethical approval was provided by the Cork University Hospital Research Ethics Committee (reference ECM (w) 07/02/2012) and all work complied with all relevant ethical regulations for working with human participants. Written informed consent was obtained from infant participant's parents. The INFANTMET cohort was subsequently followed up, with samples collected one, two and four years after birth (Supplementary data 1 and 2) from children born FT and PT (<35 weeks gestation) by CS and vaginal delivery.

**DNA extraction and processing.** Fresh faecal samples were collected from 70 infants (43% females) at year one, 57 at year two and 32 at year four. Samples were stored at 4 °C (maximum 24 h) until transfer to the laboratory, where they were placed at −80 °C until processed. DNA was extracted from a 250-mg faecal sample using a previously described modified protocol, which combined the repeat bead beating method with the QIAmp Fast DNA Stool Mini Kit (Qiagen, UK)[44,45].

**16S rRNA amplification and MiSeq sequencing.** The V4–V5 variable region of the 16S rRNA gene was amplified from 159 faecal DNA extracts. This region was chosen to allow the most accurate representation of the infant gut microbiota[10]. DNA was amplified with primers specific to the V4–V5 region of the 16S rRNA gene (Forward primer 5' TCGTCGGCAGCGTCAGATGTGTATAAGAGACAG AYTGGGYDTAAAGNG and reverse primer 5'GTCTCGTGGGCTCGGAGATG TGTATAAGAGACAGCCGTCAATTYYTTTRAGTT). Each PCR reaction contained 5 μl DNA template, 5 μl forward primer (10 μM), 5 μl reverse primer (10 μM) and 12.5 μl 2× Kapa HiFi Hotstart ready mix (Anachem, Dublin, Ireland). PCR amplification was performed as follows: 95 °C × 5 min followed by 30 cycles of 95 °C × 30 s, 65 °C × 30 s, 72 °C × 30 s, followed by 72 °C × 5 min and samples held at 4 °C. Successful PCR products were cleaned using AMPure XP magnetic bead-based purification (Labplan, Dublin, Ireland) and indexed as described in the Illumina 16S library preparation protocol. Samples were subsequently quantified and pooled in an equimolar fashion. Samples were sequenced on the MiSeq sequencing platform at Teagasc, using a 2 × 250 cycle V3 Kit, following standard Illumina sequencing protocols.

**Bioinformatics and statistical analysis.** Paired-end reads were assembled using FLASH (FLASH: fast length adjustment of short reads to improve genome assemblies). Further processing of paired-end reads including quality filtering based on a quality score of >25 and removal of mismatched barcodes and sequences was completed using QIIME version 1.9.0. Denoising, chimera detection and clustering into OTU grouping were performed using USEARCH v7[46]. OTUs were aligned using PyNAST and taxonomy was assigned using BLAST against the SILVA SSURef database release 123.

**Statistical analysis.** Statistical analysis was performed using the Calypso online software (version 8.68)[47]. All samples had >37,000 reads. Taxa present at <0.01% were removed and up to 20,000 taxa are included in the analysis, unless otherwise stated. Cumulative-sum scaling was used and data were log2 transformed to account for the non-normal distribution of taxonomic count data for alpha and beta diversity testing and repeated-measures statistical analysis[48].

Multivariate analysis including both RDA and CCA methods were used to investigate the complex associations between microbiota composition and various explanatory variables.

Alpha diversity was measured using Shannon diversity (which measures the overall diversity of a community, including the number of taxa/OTUs) and evenness (which measures how evenly abundant the taxa/OTUs are). Beta diversity was measured based on Bray–Curtis distance matrices on data from individuals sampled at one time point (years one, two or four).

Hierarchical clustering of the 100 most abundant genera ($p < 0.05$; analysis of variance) were visualised using a heatmap to determine patterns in gut microbiota based on the age of the participant. Repeated-measures statistical analysis was performed on infants sampled at ≥2 time points (Supplementary data 2). The 1000 most abundant taxa were included in the analysis. Year was chosen as the fixed effect and adjusted for gestational age at birth to determine the impact of PT birth on gut microbiota composition and dynamics. To identify the most discriminative taxa/OTUs that best characterise microbiota composition at years one, two and four, sPLS-DA was conducted using a repeated-measures design on the top 1000 most abundant genera present at ≥2 time points.

ANCOM was used to study the gut microbiota of participants at one, two and four years. ANCOM compares the log ratio of the abundance of each taxon to the abundance of all the remaining taxa one at a time and then Mann–Whitney $U$ is calculated on each log ratio. Data were not scaled or normalised prior to running ANCOM. ANCOM accounts for compositional constraints of metagenomic data to reduce false discoveries in detecting differentially abundant taxa[49].

LEfSe was used to identify predominant taxa between years one, two and four, considering biological consistency and effect relevance[50]. Calculated $p$ values are adjusted for multiple testing using the false discovery rate (FDR) correction. All $p$ values provided are following FDR correction. Significance was accepted as $p < 0.05$ following FDR correction.

**Reporting summary.** Further information on experimental design is available in the Nature Research Reporting Summary linked to this article.

## Data availability

The data that supports the findings of this study are available upon reasonable request from the corresponding author. The source data underlying Figs. 1–6 are provided as a Source Data file. We provide the raw sequencing table_tax file (Calypso V3 format) and the metadata required for each figure and used in the statistical analysis in this study.

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

## Acknowledgements

We would like to extend our most grateful thanks to all the families who agreed to take part in the INFANTMET study. The authors wish to acknowledge Dr. Paul Cotter, Dr. Fiona Crispie and Ms. Laura Finnegan from the Teagasc Sequencing facility for their role in relation to the 16S rRNA sequencing and Ms. Grace Ahern for assistance with sample processing. This work was supported by the APC Microbiome Ireland SFI funding and Government of Ireland National Development Plan by way of a Department of Agriculture, Food and Marine FIRM grant to the INFANTMET project (to C.S.) and by a grant from the Health Research Board of Ireland (HRA_-POR/2012/123) to P.W.O'T.

## Author contributions

F.F. conducted the laboratory experiments, performed the bioinformatic and statistical analysis and wrote the manuscript. C.W. conducted the laboratory experiments, performed statistical analysis and wrote the manuscript. C.J.H. was involved in sample collection and laboratory experiments and contributed to the writing of the

manuscript. C.-A.O'S. was involved in consenting participants, collection of samples and writing of the manuscript. B.N. was involved in laboratory experiments. P.W.O'T., C.A.R., E.M.D., R.P.R. and C.S. were involved in study design, interpretation of the results and writing of the manuscript. All authors read and approved the final manuscript.

## Additional information

**Competing interests:** The authors declare no competing interests.

