## [Peer Review File · Nature Communications]

Reviewers' comments:

Reviewer #1 (Remarks to the Author):

Summary:

This study include 112 individuals with gut microbiota samples taken at 1, 2, and 4 years. 16S rRNA sequencing was used to process the samples. Numerous early life exposure were examined, including gestational age at birth, delivery mode, and antibiotic usage, to assess the association with the gut microbiota alpha diversity and composition at the examined sampling times. They found no evidence that antibiotic exposure in the first year significantly shaped the later gut microbiota, but found that gestational age at birth had sustained effects, as did delivery mode.

General comments:

The writing style is somewhat technical and difficult to follow. This article could not be easily understood by someone outside of the field of microbiome research, and the paper sometimes reads like a summary of results from all of the possible analyses in a microbiome software package. The discussion does not clearly convince me of the importance or impact of these results. Furthermore, some of the details of the cohort are not clearly described, such as how many of children have 1, 2 and 3 gut microbiota samples, the details of the collected exposure data, etc. A table with cohort characteristics for each time should be included. The paper suggests that only 52 have more than one sample, but no further detail is given. It is difficult to assess the appropriateness of all of the performed analyses without better understanding the longitudinal nature of the data. There is a huge amount of temporal and inter-individual variation in early life in the gut microbiota (particularly with a wide range of perinatal exposures).

Major comments:

1. Line 6: clarify timing of antibiotic exposure
2. No mention is made of antibiotic exposure from 1-4 years. This is important information, particularly given the evidence that numerous rounds of antibiotics can have a compounding effect.
3. ANOVA is not really appropriate for compositional data, see:
Mandal, Siddhartha, et al. "Analysis of composition of microbiomes: a novel method for studying microbial composition." *Microbial ecology in health and disease* 26.1 (2015): 27663.
4. Most of the analyses performed are described by name in the text rather than conceptually. Thus, anyone unfamiliar with the specific method cannot really understand the results.

Minor comments:

1. Line 707: typo? I think three should be four.
2. Figures are of very low quality and some are illegible.
3. It is unclear how the results described in lines 79-81 are reflected in Figure 1.
4. Line 403: State version of QIIME

Reviewer #2 (Remarks to the Author):

In the paper "Microbiome Memory: Perinatal factors continue to affect the gut microbiome four years after birth" Fouhy et al describe the influence of perinatal factors on the development of the microbiome during the first years of life. The paper is written beautifully however the authors should emphasize much more it's innovation as there are several paper looking at time series of babies. My major concern with the work presented is that it seems that most of the data is not significant after corrections which I think is a problem.

Some more specific comments:

- Figure 3A is unreadable
- Line 115 – The authors should show the data as it is of interest.
- Line 132/figure 6 – what statistics were done
- Figure 6 is unreadable
- Throughout the paper in each analysis the authors take into account a different number of the most abundant OTUs. I do not think this is correct as some of the more interesting OTUs might be of low abundance. In any case the authors need to explain the different numbers in each analysis. For example, line 141 – 50 most abundant, line 165 – 300 most abundant, line 132 – 200 most abundant etc. It appears that if the authors were to use a different cutoff than no OTUs would pass FDR.
- It is unclear why the authors mention differences and then state that these differences were no longer apparent after adjustment for condition or FDR. All results that are not significant after adjusting or FDR should be removed. For example, lines 185-193, lines 172-180, line 195-203 etc.
- Line 297 change was to were.
- Line 400 – what was the range of time for storing at 4C?

- Please state the PCR protocol.
- Figures 1 and 2 can be merged to 1 figure with 2 panels
- Figure 4 – please use * to show significance
- Figure 5 is unreadable
- Fig 7 is unreadable

Response to Reviewers' comments

Reviewer #1 (Remarks to the Author):

Response: We thank reviewer 1 for all of their comments. We have taken on board all feedback and improved the manuscript accordingly. Responses to individual comments are below.

This study include 112 individuals with gut microbiota samples taken at 1, 2, and 4 years. 16S rRNA sequencing was used to process the samples. Numerous early life exposure were examined, including gestational age at birth, delivery mode, and antibiotic usage, to assess the association with the gut microbiota alpha diversity and composition at the examined sampling times. They found no evidence that antibiotic exposure in the first year significantly shaped the later gut microbiota, but found that gestational age at birth had sustained effects, as did delivery mode.

General comments:

The writing style is somewhat technical and difficult to follow. This article could not be easily understood by someone outside of the field of microbiome research, and the paper sometimes reads like a summary of results from all of the possible analyses in a microbiome software package.

Response: We have taken on board this feedback and rephrased the manuscript. We have included improved descriptions and explanations of the statistical analysis performed (lines 389-404) and we have included plain language summaries of each stats package run throughout the manuscript e.g. (105-106, 114-116, 130-131, 172-175) and the reasons for choosing the approaches used. We have reduced the technical language used and think the revised manuscript is easier to interpret.

The discussion does not clearly convince me of the importance or impact of these results.

Response: We have rewritten the discussion to expand on the importance and novelty of the results. We have included reference to relevant publications and how our research adds to existing findings (e.g. 295-296, 320-322, 300-302).

Furthermore, some of the details of the cohort are not clearly described, such as how many of children have 1, 2 and 3 gut microbiota samples, the details of the collected exposure data, etc. A table with cohort characteristics for each time should be included. The paper suggests that only 52 have more than one sample, but no further detail is given. It is difficult to assess the appropriateness of all of the performed analyses without better understanding the longitudinal nature of the data. There is a huge amount of temporal and inter-individual variation in early life in the gut microbiota (particularly with a wide range of perinatal exposures).

Response: In the revised manuscript we have further expanded on the number of samples provided by participants at the different time points (lines 90-95, 125-126, 132-133). Also SI Table2 includes information on birth mode, gestational age at birth, mother's age, antibiotics in the first year of life.

Major comments:

1. Line 6: clarify timing of antibiotic exposure

Response: This has been added to the revised manuscript (line 27)

2. No mention is made of antibiotic exposure from 1-4 years. This is important information, particularly given the evidence that numerous rounds of antibiotics can have a compounding effect.

Response: We have collected antibiotic exposure data up to one year which is provided in the revised manuscript (Supplementary information table 2, Lines 318-319, 332-335)

3. ANOVA is not really appropriate for compositional data, see:

Mandal, Siddhartha, et al. "Analysis of composition of microbiomes: a novel method for studying microbial composition." *Microbial ecology in health and disease* 26.1 (2015): 27663.

Response: Thank you for this feedback. Following consideration of this recommended text and through further investigations, we agree that ANCOM is a more suitable and robust method of statistical analysis for gut microbiota data. Therefore, we have rerun all of the statistical analysis where ANOVA was previously used and used ANCOM instead and rewritten the results accordingly. (line 394-398, 172-179). We consistently performed this analysis based on the top 1000 taxa.

4. Most of the analyses performed are described by name in the text rather than conceptually. Thus, anyone unfamiliar with the specific method cannot really understand the results.

Response: We have rephrased the manuscript and now include a more plain language explanation of the analysis performed so it is suitable for all readers (lines 103-105, 108-110, 113-115, 120-121, 12-130, 134-136, 138-139, 146-147, 180-182, 201-202)

Minor comments:

1. Line 707: typo? I think three should be four.

Response: Typo has been fixed (line 618)

2. Figures are of very low quality and some are illegible.

Response: We have remade all of the figures and they are included as separate PNG files with the revised manuscript. They have been set at 400dpi so have a very high resolution. We include 6 figures in the revised manuscript. Line 108, Fig 1A, 111 Fig 1B, Line 116 Fig 2A, Line 118 Fig 2B, 121 Fig 3A, 127 Fig 3B, 131 Fig 4, 140 Fig 5, Line 166 Fig 6A, 169 Fig 6B.

3. It is unclear how the results described in lines 79-81 are reflected in Figure 1.

Response: This was an error during editing. We have fixed the associated text to ensure it accurately reflects the revised image Fig 1A/B (Lines 104-111)

4. Line 403: State version of QIIME

Response: This has been added to the revised manuscript (line 384)

Reviewer #2 (Remarks to the Author):

Response: We thank Reviewer 2 for all of their comments and have taken on board all comments to improve our manuscript. Individual responses to comments are given below.

In the paper "Microbiome Memory: Perinatal factors continue to affect the gut microbiome four years after birth" Fouhy et al describe the influence of perinatal factors on the development of the microbiome during the first years of life. The paper is written beautifully however the authors should emphasize much more it's innovation as there are several paper looking at time series of babies.

Response: Thank you for your compliments regarding the writing style of our manuscript. We have rephrased the revised manuscript to further highlight the innovation (line 232-324, 340-341, 343-346). This study investigated whether perinatal factors including delivery mode, gestational age at birth and feeding regime would result in distinct microbial profiles extending to four years of age.

My major concern with the work presented is that it seems that most of the data is not significant after corrections which I think is a problem.

Response: In response to concerns from Reviewers 1 & 2 we have rerun the analysis using ANCOM and rewritten the associated results (line 394-397, 172-179, 202-225). This statistical analysis is a more suitable and robust statistical analysis method for gut microbiota data and has been shown to have the most sensitive FDR corrections. All of the p values in the text are the corrected p values (lines 205, 206, 176-179).

Some more specific comments:

- Figure 3A is unreadable

Response: We have remade all of the figures and they are included as separate PNG files with the revised manuscript. They have been set at 400dpi so have a very high resolution. We include 6 figures in the revised manuscript. Line 108, Fig 1A, 111 Fig 1B, Line 116 Fig 2A, Line 118 Fig 2B, 121 Fig 3A, 127 Fig 3B, 131 Fig 4, 140 Fig 5, Line 166 Fig 6A, 169 Fig 6B.

- Line 115 – The authors should show the data as it is of interest.

Response: A new figure with 6 PCoA plots has been included as SI to support the data (line 124, Supplementary information 1, Fig 2A-F).

- Line 132/figure 6 – what statistics were done

Response: The heatmap was generated using hierarchal clustering and details are given in the revised manuscript (line 138)

- Figure 6 is unreadable

Response: We have remade all of the figures and they are included as separate PNG files with the revised manuscript. They have been set at 400dpi so have a very high resolution. We include 6 figures in the revised manuscript. Line 108, Fig 1A, 111 Fig 1B, Line 116 Fig 2A, Line 118 Fig 2B, 121 Fig 3A, 127 Fig 3B, 131 Fig 4, 140 Fig 5, Line 166 Fig 6A, 169 Fig 6B.

- Throughout the paper in each analysis the authors take into account a different number of the most abundant OTUs. I do not think this is correct as some of the more interesting OTUs might be of low abundance. In any case the authors need to explain the different numbers in each analysis. For

example, line 141 – 50 most abundant, line 165 – 300 most abundant, line 132 – 200 most abundant etc. It appears that if the authors were to use a different cutoff than no OTUs would pass FDR.

Response: We have redone the analysis using 1000 taxa in all cases (lines 150, 173, 531, 546, 550). We have also included this information in all of the revised figure legends. We tested the data using much larger numbers of up to 20,000 taxa and this did not affect the results achieved. Therefore, we have now included 1000 taxa in all analysis. The exception is for the heatmap, as using greater than 100 taxa made the image too crowded and illegible (lines 138, 548). However, the pattern of clustering remained the same when we tested this analysis using 1000 taxa.

- It is unclear why the authors mention differences and then state that these differences were no longer apparent after adjustment for condition or FDR. All results that are not significant after adjusting or FDR should be removed. For example, lines 185-193, lines 172-180, line 195-203 etc.

Response: In response to concerns from Reviewers 1 & 2 we have rerun the analysis using ANCOM and rewritten the associated results (line 394-397, 172-179, 202-225). This statistical analysis is a more suitable and robust statistical analysis method for gut microbiota data and has been shown to have the most sensitive FDR corrections. All of the p values in the text are the corrected p values (lines 205, 206, 176-179).

- Line 297 change was to were.

Response: This sentence is no longer in the revised manuscript

- Line 400 – what was the range of time for storing at 4C?

Response: The samples were stored at 4°C for less than 24 hours and this has been included in the revised manuscript (line 359)

- Please state the PCR protocol.

Response: The details on the PCR protocol are provided in the revised manuscript. (lines 371-377)

- Figures 1 and 2 can be merged to 1 figure with 2 panels

Response: This has been completed in the revised manuscript (line 108, 111 Fig 1A/B)

- Figure 4 – please use * to show significance

Response: This image has been removed from the revised manuscript

- Figure 5 is unreadable

Response: We have remade all of the figures and they are included as separate PNG files with the revised manuscript. They have been set at 400dpi so have a very high resolution. This is now Fig 4 (line 131) it is a PCA plot but has been remade to improve readability

- Fig 7 is unreadable

Response: We have remade all of the figures and they are included as separate PNG files with the revised manuscript. They have been set at 400dpi so have a very high resolution. This is now Fig 6 A/B line 166, 169 and the quality has been improved and is included as a high quality file.

Reviewers' comments:

Reviewer #1 (Remarks to the Author):

This manuscript reads much more clearly after the edits, and the clarification of the details of the cohort and methods, as well as the changes in the statistical analyses, improve the strength of the manuscript.

A few additional comments:

Lines 59-60: I am not sure that there is strong evidence that gestational age is the “primary” driver of early life gut microbiota. There is evidence that it is important, but I believe that this finding that gestational age was the main driver of the gut microbiota development was among very low birth weight infants, and the uniqueness of this cohort might be very relevant to their conclusions. I think that this results should be restated and clarified.

Line 51: This sentence is unclear. Are only the FT infants born vaginally or by CS? Are all PT born via CS?

There is inconsistency in the number of significant digits reported for p-values.

The statistical methods do not describe all analyses performed. The methods section should be thorough enough for someone else to repeat the analyses. For example, the alpha and beta diversity analyses are not adequately described. The results reported in lines 120-128 seem to suggest that Adonis was used incorrectly. It is not meant for repeated measures data (I believe that `nested.anova.dbrda` allows for this), yet there is only one p-value reported across the years, suggesting that all data were included.

Line 151: What are optional study groups?

Some of the sections seem to contain material that belongs in other sections, eg. Line 153 is a method not a result.

Line 246: Is this an analysis of age of the child (1 year, 2 year, etc) or gestational age? This is unclear and not clarified by the title of the supplemental table (which did not convert clearly to PDF).

Reviewer #2 (Remarks to the Author):

I thank the reviewers for addressing my concerns. However I still feel that the innovative aspect of this paper is not highlighted enough.

Response to Reviewer's Comments

We thank both reviewer's for their comments and feedback. We have revised our manuscript accordingly and address each of their comments in detail below.

Reviewer #1 (Remarks to the Author):

This manuscript reads much more clearly after the edits, and the clarification of the details of the cohort and methods, as well as the changes in the statistical analyses, improve the strength of the manuscript.

A few additional comments:

Reviewer 1: Lines 59-60: I am not sure that there is strong evidence that gestational age is the “primary” driver of early life gut microbiota. There is evidence that it is important, but I believe that this finding that gestational age was the main driver of the gut microbiota development was among very low birth weight infants, and the uniqueness of this cohort might be very relevant to their conclusions. I think that these results should be restated and clarified.

Response: Infants can be born too early (<35 weeks), too small (<2500g) or both, therefore not all low birth weight infants are preterm. In the absence of accurate gestational age information (e.g. by first trimester ultrasound) previous studies have used birth weight as a surrogate marker of prematurity. However, in our opinion gestational age is a more accurate measure, as not all low birth weight infants are preterm (e.g. in the case of placental insufficiency and intrauterine growth restriction). For example, in our study 13% of preterm born individuals were > 2500g. The impact of birth weight on gut microbiota has now been added to **lines 58-70** and we have also highlighted the impact of other factors associated with premature delivery on gut microbiota (e.g. CS delivery). Our study found that prematurity exerts a persistent effect on the microbiome. Although we cannot predict the long term health implications of the altered microbiome in prematurity, it does give impetus for large scale studies to examine the relationship between developmental delay and the microbiome in premature infants and determining whether the persistence of these microbial signatures is related to developmental consequences.

Lines 58-70: “Worldwide approximately 10% of babies are born prematurely and up to 25% of PT survivors have adverse neurodevelopmental outcome¹³. Additionally, infants born prematurely are likely to receive antibiotic treatment which exerts significant effects on gut microbiota¹⁴. The effects of prematurity are often further confounded by CS delivery, with 43% of PT and 67% of very PT infants born by CS¹⁵. In terms of progression, patterns of microbial colonisation in the infant gut have primarily been associated with gestational age at birth, after adjusting for antibiotic exposure, mode of delivery and breastfeeding status, among others^{16, 17,18}. Indeed, many of these PT studies include very low birth weight (VLBW) infants with extended hospitalisation care, thereby exposing the juvenile microbiome to surfaces in the NICU which have previously been shown to influence gut microbiota colonisation¹⁹. Birth weight is therefore an underlying factor which plays a role in microbiota progression as VLBW infants also possess an immature immune system which influences microbe-gut interactions²⁰”,

and **lines 353-357**;

“This study has a relatively small number of PT participants (n=16); of the PTs studied, only 6% were extremely low birth weight (<1000g), 19% were very low birth weight (<1500g) and 13% were >2500g. However, we have shown that gestational age is a more accurate measure of the impact of prematurity, as not all low birth weight infants are preterm”.

Reviewer 1: Line 51: This sentence is unclear. Are only the FT infants born vaginally or by CS? Are all PT born via CS?

Response: The following sentence on **Line 50-52** has been corrected. The FT and PT infants in our INFANTMET study were born both vaginally and by CS.

“Previously, we reported on the characterisation of gut microbiota development during the first 24 weeks in a cohort of initially breastfed infants born full term (FT) and preterm (PT) (<35 weeks gestation) via vaginal delivery and Caesarean section (CS) birth mode¹⁰”.

Reviewer 1: There is inconsistency in the number of significant digits reported for p-values.

Response: We have corrected the p values in the text so all are labelled using the conventional labelling of $p<0.05$, $p<0.01$ or $p<0.001$. This has also been used for significant results in Supplementary information 2; Table 3

Reviewer 1: The statistical methods do not describe all analyses performed. The methods section should be thorough enough for someone else to repeat the analyses. For example, the alpha and beta diversity analyses are not adequately described.

Response: We have edited the text, removing the majority of the methodology descriptions from the results section and added them to the methods. The methods have been made more detailed to aid others repeating our analysis. See **lines 420-455**.

“Statistical analysis

Statistical analysis was performed using the Calypso online software (version 8.68)⁴⁸. All samples had > 37,000 reads. Taxa present at less than 0.01% were removed and up to 20,000 taxa are included in the analysis, unless otherwise stated. Cumulative-sum scaling (CSS) was used and data was log₂ transformed to account for the non-normal distribution of taxonomic count data for alpha and beta diversity testing and repeated measures statistical analysis⁴⁹.

Multivariate analysis including both RDA and CCA methods were used to investigate the complex associations between microbiota composition and various explanatory variables.

Alpha diversity was measured using Shannon diversity (which measures the overall diversity of a community, including the number of taxa/OTUs) and evenness (which measures how evenly abundant the taxa/OTUs are). Beta diversity was measured based on Bray-Curtis distance matrices on data from individuals sampled at one time point (years one, two or four).

Hierarchical clustering of the 100 most abundant genera ($p < 0.05$; ANOVA) were visualised using a heatmap to determine patterns in gut microbiota based on the age of the participant. Repeated measures statistical analysis was performed on infants sampled at two or more time points (see Supplementary information 2). The 1,000 most abundant taxa were included in the analysis. Year was chosen as the fixed effect and adjusted for gestational age at birth to determine the impact of PT birth on gut microbiota composition and dynamics.

To identify the most discriminative taxa/OTUs that best characterise microbiota composition at year one, two and four, sPLS-DA was conducted using a repeated measures design on the top 1,000 most abundant genera present at two or more time points.

Analysis of composition of microbiomes (ANCOM) was used to study the gut microbiota of participants at one, two and four years. ANCOM compares the log ratio of the abundance of each taxon to the abundance of all the remaining taxa one at a time and then the Mann-Whitney U is calculated on each log ratio. Data was not scaled or normalised prior to running ANCOM. ANCOM accounts for compositional constraints of metagenomic data to reduce false discoveries in detecting differentially abundant taxa⁵⁰.

LDA Effect Size (LEfSe) was used to identify predominant taxa between years one, two and four, considering biological consistency and effect relevance⁵¹. Calculated p values are adjusted for multiple testing using the False-Discovery-Rate (FDR) correction. All p values provided are following FDR correction. Significance was accepted as $p < 0.05$ following FDR correction”.

Reviewer 1: The results reported in lines 120-128 seem to suggest that Adonis was used incorrectly. It is not meant for repeated measures data (I believe that nested.anova.dbrda allows for this), yet there is only one p-value reported across the years, suggesting that all data were included.

Response: We thank the reviewer for bringing to our attention that Adonis is not suitable for use on repeated measures data. In the revised manuscript, we have taken the individuals studied at one timepoint (n=29 year one, n=17 year two and n=11 at year four) and re-run Adonis on these independent samples (Lines 130-133, 414-416, 579-581; Figure 3A, B, C). We can see that there is a significant effect of age on the clustering. We have also further tested to see what significance there was between these same independent individuals at years 1 and 2, 2 and 4 and 1 and 4 (updated text in the manuscript. See lines 127-134:

“Based on Bray-Curtis distance matrices on data from individuals sampled at one time point (years one, two or four) there was a notable separation based on age up to four years (Fig. 3A and 3C). The strongest significant effect was identified between years one and four, with the highest R^2 value of 0.176 and $p < 0.001$ (Fig. 3A and 3C). Gestational age at birth also has a significant impact on community composition based on OTU variance using Bray-Curtis distance matrices ($p < 0.01$) (Fig. 3B and 3C)”.

Reviewer 1: Line 151: What are optional study groups?

Response: ‘Optional study groups’ refers to the variable being taken into account i.e. what factor you are adjusting for in the statistical analysis. In this instance when we investigated the difference in gut microbiota between the years, we adjusted the data for gestational age at birth. This has been corrected on **Line 437-438**

*“Year was chosen as the fixed effect and **adjusted for** gestational age at birth to determine the impact of PT birth on gut microbiota composition and dynamics”.*

Reviewer 1: Some of the sections seem to contain material that belongs in other sections, eg. **Line 153** is a method not a result.

Response: The method highlighted on **Line 153** has been moved to the methods section (Line 402-403). We have edited the text, removing the majority of the methodology descriptions from the results section and added them to the methods. The methods have been made more detailed to aid others repeating our analysis. See **lines 420-455**, “**Statistical analysis**” mentioned above.

Reviewer 1: Line 246: Is this an analysis of age of the child (1 year, 2 year, etc) or gestational age? This is unclear and not clarified by the title of the supplemental table (which did not convert clearly to PDF).

Response: This analysis was based on the age of the child (year 1, 2, 4). In the text this has been clarified (**lines 435-438**) and also the Supplementary Table 3 has been relabelled.

“Repeated measures statistical analysis was performed on infants sampled at two or more time points (see Supplementary information 2, Supplementary Table 2). The 1,000 most abundant taxa were included in the analysis. Year was chosen as the fixed effect and adjusted for gestational age at birth to determine the impact of PT birth on gut microbiota composition and dynamics.

Reviewer #2 (Remarks to the Author):

Reviewer 2: I thank the reviewers for addressing my concerns. However I still feel that the innovative aspect of this paper is not highlighted enough.

Response: We have addressed this comment in lines 58-70, 359-369. We have highlighted our important findings, in particular that gestational age at birth still imprints on the microbiome at 4 years of age. This is extremely important as now over 10% of babies are born prematurely worldwide annually and up to 25% of preterm survivors are left with adverse neurodevelopmental outcomes. As this study is limited to correlations rather than causation we remain conservative regarding the ability to predict neurodevelopmental outcomes arising from the sustained alteration of the gut microbiome owing to preterm birth. However, our results highlight opportunities for future studies to investigate the relationship between gestational age, gut microbiota and neurodevelopmental outcomes in the light of the growing body of research on the gut-brain axis.

We have also shown that gestational age is a more accurate measure of the impact of prematurity, as not all low birth weight infants are preterm (further details are provided in response #1 - Reviewer

1).

Line 58-70: “Worldwide approximately 10% of babies are born prematurely and up to 25% of PT survivors have adverse neurodevelopmental outcome¹³. Additionally, infants born prematurely are likely to receive antibiotic treatment which exerts significant effects on gut microbiota¹⁴. The effects of prematurity are often further confounded by CS delivery, with 43% of PT and 67% of very PT infants born by CS¹⁵. In terms of progression, patterns of microbial colonisation in the infant gut have primarily been associated with gestational age at birth, after adjusting for antibiotic exposure, mode of delivery and breastfeeding status, among others^{16, 17,18}. Indeed, many of these PT studies include very low birth weight (VLBW) infants with extended hospitalisation care, thereby exposing the juvenile microbiome to surfaces in the NICU which have previously been shown to influence gut microbiota colonisation¹⁹. Birth weight is therefore an underlying factor which plays a role in microbiota progression as VLBW infants also possess an immature immune system which influences microbe-gut interactions²⁰”,

and lines 359-369:

“The results of this study have shown that gestational age at birth still imprints on the microbiome at four years of age. This is extremely important as now worldwide over 10% of babies are born prematurely annually and up to 25% of preterm survivors are left with adverse neurodevelopmental outcomes. This study is limited to correlations rather than causation between prematurity and gut microbiota development and thus we cannot predict the implications on neurodevelopmental outcomes of the altered microbiome in prematurity. However, our findings of a persistent impact of gestational age on gut microbiota highlights the opportunities for large scale studies to investigate the relationship between gestational age, gut microbiota and the gut-brain axis. Moreover, this study opens up opportunities whereby live biotherapeutic interventions may be beneficial in preterm infants to modulate the gut microbiota to resemble their full term born counterparts.”

REVIEWERS' COMMENTS:

Reviewer #1 (Remarks to the Author):

Thank you for addressing all of the comments.

I particularly like the emphasis on lines 359-369 about how this study is limited to correlations.